# A Scheme for Controlled Cyclic Asymmetric Remote State Preparation in Noisy Environment

**Nan Zhao** * , **Tingting Wu** , **Yan Yu** and **Changxing Pei**

State Key Laboratory of Integrated Services Networks, Xidian University, Taibai South Road, Electronic City Street, Yanta District, Xi'an 710071, China; wutintin2016@outlook.com (T.W.); yuyan610@126.com (Y.Y.); chxpei@xidian.edu.cn (C.P.)
* Correspondence: zhaonan@xidian.edu.cn

**Abstract:** As research on quantum computers and quantum information transmission deepens, the multi-particle and multi-mode quantum information transmission has been attracting increasing attention. For scenarios where multi-parties transmit sequentially increasing qubits, we put forward a novel (N + 1)-party cyclic remote state preparation (RSP) protocol among an arbitrary number of players and a controller. Specifically, we employ a four-party scheme in the case of a cyclic asymmetric remote state preparation scheme and demonstrate the feasibility of the scheme on the IBM Quantum Experience platform. Furthermore, we present a general quantum channel expression under different circulation directions based on the n-party. In addition, considering the impact of the actual environment in the scheme, we discuss the feasibility of the scheme affected by different noises.

**Keywords:** controlled cyclic asymmetric; qubits sequentially increasing; IBM quantum experience; bit-phase flip noise; decoherence rate; coefficient of the desired state

## 1. Introduction

During the past few years, quantum communication theory and technology have been further extended. Quantum entanglement is a crucial resource in quantum information processing tasks, and has been widely applied in various fields such as quantum teleportation (QT), remote state preparation (RSP), quantum key distribution (QKD) and quantum secret sharing (QSS). In 1982, Aspect successfully confirmed the phenomenon of quantum entanglement through experiments [1]. In 1993, Bennett proposed the concept of QT [2], which has great potential for protecting sensitive information and accelerating classical computation. Then, Lo [3] first introduced remote state preparation (RSP) at the beginning of the 21st century. In RSP [4–6], Alice can remotely prepare an arbitrary single-qubit quantum state for Bob by using a classical bit communication and a shared entangled state. Besides, Alice knows all the information of the single-qubit quantum state to be prepared by Bob, and Alice does not have to hold the single-qubit quantum state. Since then, the novel type of RSP has drawn considerable attention, and various related schemes [7–11], like CRSP (controlled RSP) [12,13], CJRSP (controlled joint) [14] and CBRSP (controlled bidirectional RSP), have been presented. In addition, with the development of entanglement source preparation, the scheme of RSP has been expanded from single qubit to multiple qubits [15]. In 2016, Chen et al. pointed out an efficient scheme for the RSP of an arbitrary five-qubit Brown-type state [16]. Binayak and Ding introduced the joint remote preparation of an arbitrary six-qubit cluster-type state [17,18]. In 2018, Wu and Fang discussed the bidirectional and hybrid quantum information transmission schemes [19,20]. A JRSP scheme of the arbitrary eight-qubit cluster-type state was put forward in 2020 [21]. However, we note that almost all the protocols mentioned above are only considered in one-way directional or bidirectional preparations. In fact, the information network is made up of two or more nodes, that is, a large information system usually is composed by some subsystems. Naturally, some cyclic RSP schemes [22,23] are investigated, information

switching is not located in two-party system. In 2019, Li [24] et al. discussed the scheme of the controlled cyclic quantum teleportation of an arbitrary two-qubit entangled state by employing a ten-qubit entangled state as a quantum channel. Sang [25] et al. explored a scheme of four-party cyclic RSP of an arbitrary single-qubit state by employing a ten-qubit maximally entangled state as the quantum channel. In quantum communication, the remote preparation of multi-particle quantum states between multi-party is inevitable. So we put forward the cyclic asymmetric remote preparation scheme with qubits sequentially increasing, which is a novel transmission mode of cyclic RSP. Furthermore, we generalize the scheme to send multiple qubits quantum information among multiple players in an asymmetric manner.

The influence of decoherence caused by the actual environment has always been the major factor affecting the effect of quantum state transmission; it is meaningful to investigate the cyclic controlled RSP in a noisy environment. In recent years, some RSP schemes that operated in noisy environments have been proposed [26–29]. In 2018, Sun [30] put forward an asymmetrically controlled two-way joint RSP in a noisy environment for the preparation of single and three equatorial states. The deterministic hierarchical remote state preparation of a two-qubit entangled state employing a Brown state in a noisy environment was proposed in 2020 [31].

Since 2016, IBM Quantum Experience has allowed players to design quantum circuits employing an interactive graphical user interface, test and simulate these circuits on classic computers and actual quantum processors. The actual relevance of an optimized scheme lies in the experimental realization of the scheme. The different kinds of theoretical protocols for quantum communication and computation [32–34] have already been verified in the IBM quantum experience. IBM Quantum Experience has been utilized to perform many practical experiments on quantum chips, including quantum simulation, the development of quantum algorithms [35], testing quantum information theory tasks, quantum cryptography, quantum error correction [36], quantum applications [37], and so forth.

In this paper, we put forward a novel (N + 1)-party cyclic RSP protocol for preparing sequentially increasing qubits among arbitrary number of players with a controller. To be specific, the i-th party prepare an i-qubit state for the (i + 1)-th party in a loop of N(1, 2, ... i, i + 1, ... n)-party, and the n-th party prepares the n-qubit state for the first party. Moreover, we demonstrate the special case of our scheme on the IBM quantum computer by designing appropriate quantum circuits using single-qubit and two-qubit quantum gates. For verifying the practicality of our scheme, we also discuss the variation of the fidelity of the scheme affected by different noises.

This paper is structured as follows—in Section 2, we specifically introduce a controlled cyclic asymmetric RSP scheme in which Alice remotely prepares an arbitrary single-qubit state for Bob, Bob prepares an arbitrary two-qubit state to Charlie, and Charlie prepares an arbitrary three-qubit state for Alice. Then, we construct the quantum circuit simulation of this scheme in IBM's quantum simulator and analyse the mean probability of the output results. Furthermore, we generalize the scheme to (N + 1)-party controlled cyclic RSP protocol for sequentially increasing qubits in Section 3. In Section 4, we analyze and compare the fidelity of output state in four noisy environments (depolarization noise, amplitude damping noise, phase damping noise, and bit-phase flip noise). Finally, we give a brief comparison and conclusion of the scheme in Section 5.

## 2. Four-Party Controlled Cyclic Asymmetrical RSP Protocol of Sequentially Increasing Qubits States

In this section, we aim at a situation where the amount of information transmitted between multiple participants is sequentially increasing. Suppose that there are four participants, employing ten quantum entangled states as quantum channels. Alice prepares any single-qubit state $|\xi_A\rangle = a_0 e^{i\theta_0}|0\rangle + a_1 e^{i\theta_1}|1\rangle$ for Bob, Bob prepares any two-qubit state $|\xi_B\rangle = b_0 e^{i\phi_0}|00\rangle + b_1 e^{i\phi_1}|11\rangle$ for Charlie, Charlie prepares any three-qubit state $|\xi_C\rangle = c_0 e^{i\gamma_0}|000\rangle + c_1 e^{i\gamma_1}|111\rangle$ for Alice, and David is the controller, where $a_0, a_1, b_0, b_1, c_0, c_1 \in R$,

$a_0^2 + a_1^2 = 1$, $b_0^2 + b_1^2 = 1$, $c_0^2 + c_1^2 = 1$, $\theta_0, \theta_1, \phi_0, \phi_1, \gamma_0, \gamma_1 \in [0, 2\pi]$. The scenario of the scheme is shown in Figure 1.

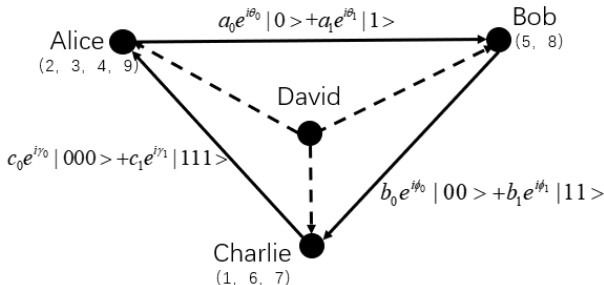

**Figure 1.** The schematic of the four-party cyclic asymmetric remote state preparation (RSP) protocol. The black arrow points out the direction of the communications and the dashed line represents the control information.

The quantum channel linking the Alice, Bob, Charlie and David has the form

$$|\varphi\rangle_{12345678910}$$
$$= \tfrac{1}{4}[(|0000\rangle + |1111\rangle)_{1234}(|000\rangle + |111\rangle)_{567}(|00\rangle + |11\rangle)_{89}|0\rangle_{10} \tag{1}$$
$$+(|0011\rangle - |1100\rangle)_{1234}(|011\rangle - |100\rangle)_{567}(|01\rangle - |10\rangle)_{89}|1\rangle_{10}],$$

where Alice holds qubits (2,3,4,9), Bob holds qubits (5,8), Charlie holds qubits (1,6,7), and David holds qubit 10.

### 2.1. Preparation of Quantum Channel

The quantum channel in the simplest case is a ten-qubits entangled state shared by the sender Alice, Bob, Charlie and the receiver David, which can be realized by Hadamard (H), Controlled-NOT (CNOT) operations and Pauli X, Y, Z gates in "IBM qasm Simulator".

Firstly, the product state of ten-qubit states initialized to $|0\rangle$ is used as the input state $|\varphi_0\rangle$ in the quantum circuit, which can be expressed as:

$$|\varphi_0\rangle = |0\rangle_1 \otimes |0\rangle_2 \cdots \otimes |0\rangle_{10} \tag{2}$$

Then, performing H gates on qubits (1, 5, 8, 10), and then the input state $|\varphi_0\rangle$ is converted to $|\varphi_1\rangle$ as follows:

$$|\varphi_1\rangle = (|0000\rangle + |1000\rangle)_{1234} \otimes (|000\rangle + |100\rangle)_{567} \otimes (|00\rangle + |10\rangle)_{89}(|0\rangle + |1\rangle)_{10}. \tag{3}$$

Several CNOT operations are performed on the following qubit pairs (1,2), (1,3), (1,4), (5,6), (5,7), (8,9). When qubit 10 is $|1\rangle$, qubits 3, 4, 6, 7, and 9 are performed Pauli X gate operation, and qubits 1, 5, and 8 are performed Pauli Z gate operation. Afterwards, the quantum channel is reconstructed as Formula (1).

Then, Alice, Bob and Charlie respectively performing CNOT operation on qubits pairs $(9, a')$, $(5, b'')$, $(1, c''')$, qubits 9, 5, 1 control auxiliary qubits $a'$ and $b''$, $c'''$. We get the thirteen-qubits target channel and it can be described as follows:

$$\left|\varphi'\right\rangle_{12345678910a'b''c'''}$$
$$= \tfrac{1}{4}[(|0000\rangle|0\rangle_{c'''} + |1111\rangle|1\rangle_{c'''})_{1234}(|000\rangle|0\rangle_{b''} + |111\rangle|1\rangle_{b''})_{567} \tag{4}$$
$$(|00\rangle|0\rangle_{a'} + |11\rangle|1\rangle_{a'})_{89}|0\rangle_{10} + (|0011\rangle|0\rangle_{c'''} - |1100\rangle|1\rangle_{c'''})_{1234}$$
$$(|011\rangle|0\rangle_{b''} - |100\rangle|1\rangle_{b''})_{567}|01\rangle|1\rangle_{a'} - |10\rangle|0\rangle_{a'})_{89}|1\rangle_{10}].$$

The process and the result of the construction of the target channel in IBM qasm Simulator is illustrated by Figure 2.

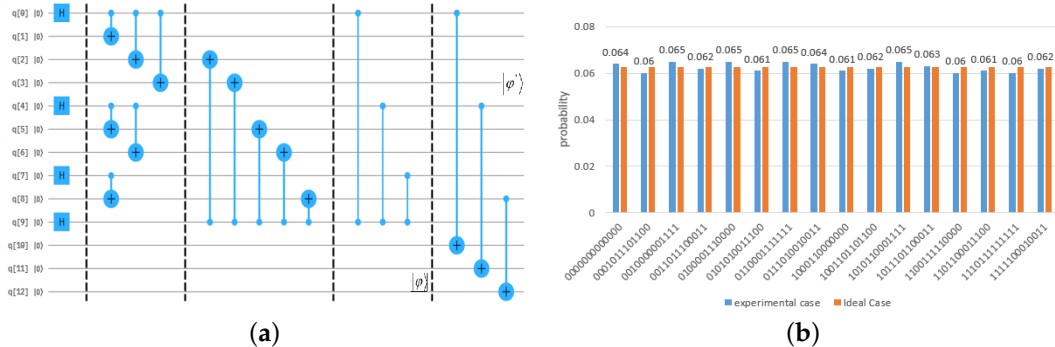

(a)                                                    (b)

**Figure 2.** The construction of the target channel. (**a**) Quantum circuit illustrating the construction of target channel. (**b**) The histogram shows theoretical and experimental results of the quantum state's mean probability distribution. The x-axis of this histogram is arranged in the order of $(c''', b'', \cdots 2, 1)$ from the bottom to the top, showing the average probability and percentage error of each state in the target channel of 13 qubits.

Theoretically, the distribution probability of each quantum state in the target channel is 0.0625 in ideal case. Based on the Figure 2b, we calculate that the standard deviation of the average probability distribution is 0.002. The diversity in mean probability and ideal case is due to the decoherence effects and gate errors.

### 2.2. Description of Four-Party Controlled Cyclic Asymmetrical RSP Protocol

In this scheme, the four participants including three players and one controller, and each of the players acts both as a sender and as a receiver. From the schematic given in Figure 3, the procedure of the four-party controlled cyclic asymmetrical RSP protocol is detailed as follows:

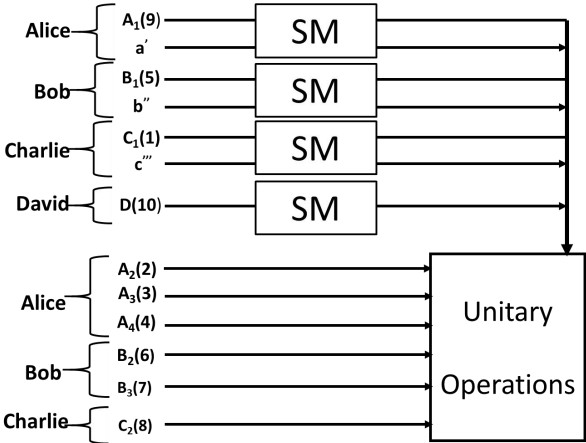

**Figure 3.** Diagram of four-party controlled cyclic asymmetrical RSP protocol, where SM is short for single-qubit measurement.

**Step 1**: Alice selects an appropriate measurement basis according to the known quantum state to perform a single qubit measurement on his qubits. After completing the measurement operations, Alice announces his outcomes to the Bob via the classical channels.

Alice performs a single qubit measurement on qubit 9 based on the following basis:

$$\begin{cases} |\Phi^+\rangle_9 = (a_0|0\rangle + a_1|1\rangle)_9 \\ |\Phi^-\rangle_9 = (a_1|0\rangle - a_0|1\rangle)_9 \end{cases} \tag{5}$$

The measurement result has two possibilities, and the remaining qubits collapse with a probability of 0.5 to a new state as follows:

$$
{}_9\langle\Phi^{\pm}\mid\varphi'\rangle_{123456789,10,a'b''c'''}
\begin{cases}
= \frac{1}{4}[(|0000\rangle|0\rangle_{c'''} + |1111\rangle|1\rangle_{c'''})_{1234}(|000\rangle|0\rangle_{b''} \\
+ |111\rangle|1\rangle_{b''})_{567}(a_0|0\rangle_8|0\rangle_{a'} + a_1|1\rangle_8|1\rangle_{a'})|0\rangle_{10} \\
+ (|0011\rangle|0\rangle_{c'''} - |1100\rangle|1\rangle_{c'''})_{1234}(|011\rangle|0\rangle_{b''} \\
- |100\rangle|1\rangle_{b''})_{567}(a_1|0\rangle_8|1\rangle_{a'} - a_0|1\rangle_8|0\rangle_{a'})|1\rangle_{10}] \\
= \frac{1}{4}[(|0000\rangle|0\rangle_{c'''} + |1111\rangle|1\rangle_{c'''})_{1234}(|000\rangle|0\rangle_{b''} \\
+ |111\rangle|1\rangle_{b''})_{567}(a_1|0\rangle_8|0\rangle_{a'} - a_0|1\rangle_8|1\rangle_{a'})|0\rangle_{10} \\
+ (|0011\rangle|0\rangle_{c'''} - |1100\rangle|1\rangle_{c'''})_{1234}(|011\rangle|0\rangle_{b''} \\
- |100\rangle|1\rangle_{b''})_{567}(-a_0|0\rangle_8|1\rangle_{a'} - a_1|1\rangle_8|0\rangle_{a'})|1\rangle_{10}].
\end{cases}
\tag{6}
$$

Alice performs a single qubit measurement on an auxiliary qubit $a'$.

For ${}_9\langle\Phi^{+}|\varphi'\rangle_{123456789,10,a'b''c'''}$, the basis are

$$
\begin{cases}
|\Psi^{+}\rangle_{a'} = \frac{1}{\sqrt{2}}(e^{-i\theta_0}|0\rangle + e^{-i\theta_1}|1\rangle)_{a'} \\
|\Psi^{-}\rangle_{a'} = \frac{1}{\sqrt{2}}(e^{-i\theta_0}|0\rangle - e^{-i\theta_1}|1\rangle)_{a'}.
\end{cases}
\tag{7}
$$

The remaining qubits collapse with a probability of 0.5 to a new state as follows:

$$
{}_{a'}\langle\Psi^{\pm}|{}_9\langle\Phi^{+}\mid\varphi'\rangle_{123456789,10,a'b''c'''}
\begin{cases}
= \frac{1}{4\sqrt{2}}[(|0000\rangle|0\rangle_{c'''} + |1111\rangle|1\rangle_{c'''})_{1234}(|000\rangle|0\rangle_{b''} \\
+ |111\rangle|0000\rangle_{b''})_{567}(a_0e^{i\theta_0}|0\rangle_8 + a_1e^{i\theta_1}|1\rangle_8)|0\rangle_{10} \\
+ (|0011\rangle|0\rangle_{c'''} - |1100\rangle|1\rangle_{c'''})_{1234}(|011\rangle|0\rangle_{b''} \\
- |100\rangle|1\rangle_{b''})_{567}(a_1e^{i\theta_1}|0\rangle_8 - a_0e^{i\theta_0}|1\rangle_8)|1\rangle_{10}] \\
= \frac{1}{4\sqrt{2}}[(|0000\rangle|0\rangle_{c'''} + |1111\rangle|1\rangle_{c'''})_{1234}(|000\rangle|0\rangle_{b''} \\
+ |111\rangle|1\rangle_{b''})_{567}(a_0e^{i\theta_0}|0\rangle_8 - a_1e^{i\theta_1}|1\rangle_8)|0\rangle_{10} \\
+ (|0011\rangle|0\rangle_{c'''} - |1100\rangle|1\rangle_{c'''})_{1234}(|011\rangle|0\rangle_{b''} \\
- |100\rangle|1\rangle_{b''})_{567}(-a_1e^{i\theta_1}|0\rangle_8 - a_0e^{i\theta_0}|1\rangle_8)|1\rangle_{10}].
\end{cases}
\tag{8}
$$

For ${}_9\langle\Phi^{-}\mid\varphi'\rangle_{12345678910a'b''c'''}$, the basis are

$$
\begin{cases}
|\Omega^{+}\rangle_{a'} = \frac{1}{\sqrt{2}}(e^{-i\theta_1}|0\rangle + e^{-i\theta_0}|1\rangle)_{a'} \\
|\Omega^{-}\rangle_{a'} = \frac{1}{\sqrt{2}}(e^{-i\theta_1}|0\rangle - e^{-i\theta_0}|1\rangle)_{a'}.
\end{cases}
\tag{9}
$$

The remaining qubits collapse with a probability of 0.5 to a new state as follows:

$$
{}_{a'}\langle\Omega^{\pm}|{}_9\langle\Phi^{-}\mid\varphi'\rangle_{123456789,10,a'b''c'''}
$$
$$
\begin{cases}
= \frac{1}{4\sqrt{2}}[(|0000\rangle|0\rangle_{c'''} + |1111\rangle|1\rangle_{c'''})_{1234}(|000\rangle|0\rangle_{b''} \\
+ |111\rangle|1\rangle_{b''})_{567}(a_1e^{i\theta_1}|0\rangle_8 - a_0e^{i\theta_0}|1\rangle_8)|0\rangle_{10} \\
+ (|0011\rangle|0\rangle_{c'''} - |1100\rangle|1\rangle_{c'''})_{1234}(|011\rangle|0\rangle_{b''} \\
- |100\rangle|1\rangle_{b''})_{567}(-a_0e^{i\theta_0}|0\rangle_8 - a_1e^{i\theta_1}|1\rangle_8)|1\rangle_{10}] \\
= \frac{1}{4\sqrt{2}}[(|0000\rangle|0\rangle_{c'''} + |1111\rangle|1\rangle_{c'''})_{1234}(|000\rangle|0\rangle_{b''} \\
+ |111\rangle|1\rangle_{b''})_{567}(a_1e^{i\theta_1}|0\rangle_8 + a_0e^{i\theta_0}|1\rangle_8)|1\rangle_{10} \\
+ (|0011\rangle|0\rangle_{c'''} - |1100\rangle|1\rangle_{c'''})_{1234}(|011\rangle|0\rangle_{b''} \\
- |100\rangle|1\rangle_{b''})_{567}(a_0e^{i\theta_0}|0\rangle_8 - a_1e^{i\theta_1}|1\rangle_8)|1\rangle_{10}].
\end{cases}
\tag{10}
$$

**Step 2:** Bob selects an appropriate measurement basis according to the known quantum state to perform a single qubit measurement on his qubits. After completing the measurement, Bob announces his outcomes to the Charlie via the classical channels.

Bob performs a single qubit measurement on qubit 5 based on the following basis:

$$\begin{cases} |\Phi^+\rangle_5 = (b_0|0\rangle + b_1|1\rangle)_5 \\ |\Phi^-\rangle_5 = (b_1|0\rangle - b_0|1\rangle)_5. \end{cases} \tag{11}$$

The measurement result has eight possibilities, and the remaining qubits collapse with a probability of 0.125 to a new state ${}_5\langle\Phi^\pm|_{a'}\langle\Psi^\pm|_9\langle\Phi^+ \mid \varphi'\rangle_{123456789,10,a'b''c'''}$ ${}_5\langle\Phi^\pm|_{a'}\langle\Omega^\pm|_9\langle\Phi^- \mid \varphi'\rangle_{123456789,10,a'b''c'''}$.

Then Bob performs a single qubit measurement on an auxiliary qubit $b''$.

For ${}_5\langle\Phi^+|_{a'}\langle\Psi^\pm|_9\langle\Phi^+ \mid \varphi'\rangle_{123456789,10,a'b''c'''}$ ${}_5\langle\Phi^+|_{a'}\langle\Omega^\pm|_9\langle\Phi^- \mid \varphi'\rangle_{123456789,10,a'b''c'''}$, the basis are

$$\begin{cases} |\Psi^+\rangle_{b''} = \frac{1}{\sqrt{2}}(e^{-i\phi_0}|0\rangle + e^{-i\phi_1}|1\rangle)_{b''} \\ |\Psi^-\rangle_{b''} = \frac{1}{\sqrt{2}}(e^{-i\phi_0}|0\rangle - e^{-i\phi_1}|1\rangle)_{b''}. \end{cases} \tag{12}$$

If Bob's single-qubit projection measurement result is $|\Phi^+\rangle_5$, $|\Psi^+\rangle_{b''}$, the remaining qubits collapse with a probability of 0.25 to a new state as follows:

$${}_{b''}\langle\Psi^+|_5\langle\Phi^+|_{a'}\langle\Psi^\pm|_9\langle\Phi^+ \mid \varphi'\rangle_{123456789,10,a'b''c'''}$$

$$\begin{cases} = \frac{1}{8}[(|0000\rangle|0\rangle_{c'''} + |1111\rangle|1\rangle_{c'''})_{1234}(b_0 e^{i\phi_0}|00\rangle_{67} + b_1 e^{i\phi_1}|11\rangle_{67}) \\ (a_0 e^{i\theta_0}|0\rangle_8 + a_1 e^{i\theta_1}|1\rangle_8)|0\rangle_{10} + (|0011\rangle|0\rangle_{c'''} - |1100\rangle|1\rangle_{c'''})_{1234} \\ (b_0 e^{i\phi_0}|11\rangle_{67} - b_1 e^{i\phi_1}|00\rangle_{67})(a_1 e^{i\theta_1}|0\rangle_8 - a_0 e^{i\theta_0}|1\rangle_8)|1\rangle_{10}] \\ = \frac{1}{8}[(|0000\rangle|0\rangle_{c'''} + |1111\rangle|1\rangle_{c'''})_{1234}(b_0 e^{i\phi_0}|00\rangle_{67} + b_1 e^{i\phi_1}|11\rangle_{67}) \\ (a_0 e^{i\theta_0}|0\rangle_8 - a_1 e^{i\theta_1}|1\rangle_8)|0\rangle_{10} + (|0011\rangle|0\rangle_{c'''} - |1100\rangle|1\rangle_{c'''})_{1234} \\ (b_0 e^{i\phi_0}|11\rangle_{67} - b_1 e^{i\phi_1}|00\rangle_{67})(-a_1 e^{i\theta_1}|0\rangle_8 - a_0 e^{i\theta_0}|1\rangle_8)|1\rangle_{10}] \end{cases} \tag{13}$$

$${}_{b''}\langle\Psi^+|_5\langle\Phi^+|_{a'}\langle\Omega^\pm|_9\langle\Phi^- \mid \varphi'\rangle_{123456789,10,a'b''c'''}$$

$$\begin{cases} = \frac{1}{8}[(|0000\rangle|0\rangle_{c'''} + |1111\rangle|1\rangle_{c'''})_{1234}(b_0 e^{i\phi_0}|00\rangle_{67} + b_1 e^{i\phi_1}|11\rangle_{67}) \\ (a_1 e^{i\theta_1}|0\rangle_8 - a_0 e^{i\theta_0}|1\rangle_8)|0\rangle_{10} + (|0011\rangle|0\rangle_{c'''} - |1100\rangle|1\rangle_{c'''})_{1234} \\ (b_0 e^{i\phi_0}|11\rangle_{67} - b_1 e^{i\phi_1}|00\rangle_{67})(-a_0 e^{i\theta_0}|0\rangle_8 - a_1 e^{i\theta_1}|1\rangle_8)|1\rangle_{10}] \\ = \frac{1}{8}[(|0000\rangle|0\rangle_{c'''} + |1111\rangle|1\rangle_{c'''})_{1234}(b_0 e^{i\phi_0}|00\rangle_{67} + b_1 e^{i\phi_1}|11\rangle_{67}) \\ (a_1 e^{i\theta_1}|0\rangle_8 + a_0 e^{i\theta_0}|1\rangle_8)|0\rangle_{10} + (|0011\rangle|0\rangle_{c'''} - |1100\rangle|1\rangle_{c'''})_{1234} \\ (b_0 e^{i\phi_0}|11\rangle_{67} - b_1 e^{i\phi_1}|00\rangle_{67})(a_0 e^{i\theta_0}|0\rangle_8 - a_1 e^{i\theta_1}|1\rangle_8)|1\rangle_{10}]. \end{cases} \tag{14}$$

For ${}_5\langle\Phi^-|_{a'}\langle\Psi^\pm|_9\langle\Phi^+ \mid \varphi'\rangle_{123456789,10,a'b''c'''}$ ${}_5\langle\Phi^-|_{a'}\langle\Omega^\pm|_9\langle\Phi^- \mid \varphi'\rangle_{123456789,10,a'b''c'''}$, the basis are

$$\begin{cases} |\Omega^+\rangle_{b''} = \frac{1}{\sqrt{2}}(e^{-i\phi_1}|0\rangle + e^{-i\phi_0}|1\rangle)_{a'} \\ |\Omega^-\rangle_{b''} = \frac{1}{\sqrt{2}}(e^{-i\phi_1}|0\rangle - e^{-i\phi_0}|1\rangle)_{a'}. \end{cases} \tag{15}$$

If Bob's single-qubit projection measurement result is $|\Omega^+>_{b''}$ $|\Phi^-\rangle_5$, the remaining qubits collapse with a probability of 0.25 to a new state as follows:

$${}_{b''}\langle\Omega^+|_5\langle\Phi^-|_{a'}\langle\Psi^\pm|_9\langle\Phi^+ \mid \varphi'\rangle_{123456789,10,a'b''c'''}$$

$$\begin{cases} = \frac{1}{8}[(|0000\rangle|0\rangle_{c'''} + |1111\rangle|1\rangle_{c'''})_{1234}(b_1 e^{i\phi^1}|00\rangle_{67} - b_0 e^{i\phi^0}|11\rangle_{67}) \\ (a_0 e^{i\theta_0}|0\rangle_8 + a_1 e^{i\theta_1}|1\rangle_8)|0\rangle_{10} + (|0011\rangle|0\rangle_{c'''} - |1100\rangle|1\rangle_{c'''})_{1234} \\ (b_1 e^{i\phi_1}|11\rangle_{67} + b_0 e^{i\phi_0}|00\rangle_{67})(a_1 e^{i\theta_1}|0\rangle_8 - a_0 e^{i\theta_0}|1\rangle_8)|1\rangle_{10}] \\ = \frac{1}{8}[(|0000\rangle|0\rangle_{c'''} + |1111\rangle|1\rangle_{c'''})_{1234}(b_1 e^{i\phi_1}|00\rangle_{67} - b_0 e^{i\phi_0}|11\rangle_{67}) \\ (a_0 e^{i\theta_0}|0\rangle_8 - a_1 e^{i\theta_1}|1\rangle_8)|0\rangle_{10} + (|0011\rangle|0\rangle_{c'''} - |1100\rangle|1\rangle_{c'''})_{1234} \\ (b_1 e^{i\phi_1}|11\rangle_{67} + b_0 e^{i\phi_0}|00\rangle_{67})(-a_1 e^{i\theta_1}|0\rangle_8 - a_0 e^{i\theta_0}|1\rangle_8)|1\rangle_{10}] \end{cases} \tag{16}$$

$$
\begin{aligned}
&_{b''}\langle\Omega^+|_5\langle\Phi^-|_{a'}\langle\Omega^\pm|_9\Big\langle\Phi^-\ \Big|\ \varphi'\Big\rangle_{123456789,10,a'b''c'''}\\
&\left\{
\begin{aligned}
&= \tfrac{1}{8}[(|0000\rangle|0\rangle_{c'''}+|1111\rangle|1\rangle_{c'''})_{1234}(b_1e^{i\phi^1}|00\rangle_{67}-b_0e^{i\phi^0}|11\rangle_{67})\\
&(a_1e^{i\theta_1}|0\rangle_8-a_0e^{i\theta_0}|1\rangle_8)|0\rangle_{10}+(|0011\rangle|0\rangle_{c'''}-|1100\rangle|1\rangle_{c'''})_{1234}\\
&(b_1e^{i\phi_1}|00\rangle_{67}+b_0e^{i\phi_0}|11\rangle_{67})(-a_0e^{i\theta_0}|0\rangle_8-a_1e^{i\theta_1}|1\rangle_8)|1\rangle_{10}]\\
&= \tfrac{1}{8}[(|0000\rangle|0\rangle_{c'''}+|1111\rangle|1\rangle_{c'''})_{1234}(b_1e^{i\phi_1}|00\rangle_{67}-b_0e^{i\phi_0}|11\rangle_{67})\\
&(a_1e^{i\theta_1}|0\rangle_8+a_0e^{i\theta_0}|1\rangle_8)|0\rangle_{10}+(|0011\rangle|0\rangle_{c'''}-|1100\rangle|1\rangle_{c'''})_{1234}\\
&(b_1e^{i\phi_1}|00\rangle_{67}+b_0e^{i\phi_0}|11\rangle_{67})(a_0e^{i\theta_0}|0\rangle_8-a_1e^{i\theta_1}|1\rangle_8)|1\rangle_{10}].
\end{aligned}
\right.
\end{aligned}
\tag{17}
$$

**Step 3:** Charlie selects an appropriate measurement basis according to the known quantum state to perform a single qubit measurement on his qubits. After completing the measurement, Charlie announces his outcomes to the Alice via the classical channels.

Charlie performs a single qubit measurement on qubit 1 based on the following basis:

$$
\begin{cases}
|\Phi^+\rangle_1 = (c_0|0\rangle + c_1|1\rangle)_1\\
|\Phi^-\rangle_1 = (c_1|0\rangle - c_0|1\rangle)_1.
\end{cases}
\tag{18}
$$

Then Charlie performs a single qubit measurement on an auxiliary qubit $c'''$.

If Charlie performs a single qubit measurement on qubit 1 based on the $|\Phi^+\rangle_1$, the basis are

$$
\begin{cases}
|\Psi^+\rangle_{c'''} = \tfrac{1}{\sqrt{2}}(e^{-i\gamma_0}|0\rangle + e^{-i\gamma_1}|1\rangle)_{a'}\\
|\Psi^-\rangle_{c'''} = \tfrac{1}{\sqrt{2}}(e^{-i\gamma_0}|0\rangle - e^{-i\gamma_1}|1\rangle)_{a'}.
\end{cases}
\tag{19}
$$

If Charlie performs a single qubit measurement on qubit 1 based on the $|\Phi^-\rangle_1$, the basis are

$$
\begin{cases}
|\Omega^+\rangle_{c'''} = \tfrac{1}{\sqrt{2}}(e^{-i\gamma_1}|0\rangle + e^{-i\gamma_0}|1\rangle)_{a'}\\
|\Omega^-\rangle_{c'''} = \tfrac{1}{\sqrt{2}}(e^{-i\gamma_1}|0\rangle - e^{-i\gamma_0}|1\rangle)_{a'}.
\end{cases}
\tag{20}
$$

**Step 4**: David carries out a single qubit measurement on qubit 10 based on the following basis: $|0\rangle, |1\rangle$.

With the help of the controller David, the senders Alice, Bob and Charlie can exchange their desired quantum state at the same time. The final result of the controlling party David measuring particle 10 makes the participants Alice, Bob, and Charlie must carry out corresponding unitary transformations, respectively, to obtain all the information of the quantum state. The controller ensures the security of the quantum communication scheme and provides robustness against complex eavesdropping attacks.

**Step 5: The unitary transformations**

If the controller, David's, basis is $|0\rangle$, then the qubits 2,3,4,6,7 and 8 will collapse into the following product state $|\varphi\rangle_{234678} = (c_0e^{i\gamma_0}|000\rangle + c_1e^{i\gamma_1}|111\rangle) \otimes (b_0e^{i\phi_0}|00\rangle + b_1e^{i\phi_1}|11\rangle) \otimes (a_0e^{i\theta_0}|0\rangle + a_1e^{i\theta_1}|1\rangle)$. Alice, Bob and Charlie carry out a corresponding unitary operation $I^{234} \otimes I^{67} \otimes I^8$ to obtain the desired quantum state respectively.

If the controller David's basis is $|1\rangle$, then the qubits 2,3,4,6,7 and 8 will collapse into the following product state $|\varphi\rangle_{234678} = (c_0e^{i\gamma_0}|011\rangle - c_1e^{i\gamma_1}|100\rangle) \otimes (b_0e^{i\phi_0}|11\rangle - b_1e^{i\phi_1}|00\rangle) \otimes (a_1e^{i\theta_1}|0\rangle - a_0e^{i\theta_0}|1\rangle)$. Alice, Bob and Charlie carry out a corresponding unitary operation $\sigma_z^2 \otimes \sigma_x^{34} \otimes i\sigma_y^6 \otimes \sigma_x^7 \otimes -i\sigma_y^8$ to obtain the desired quantum state, respectively.

For the other 127 measurement results, Alice, Bob and Charlie can perform an appropriate unitary operation on their qubit according to the single-qubit projection measurement result, and then reconstruct the desired quantum state for 100%. Eight situations of the possible outcomes of the measurement results and corresponding unitary operations are illustrated in the following Table 1.

**Table 1.** Joint collapsed states of some cases and reconstructed unitary transformation.

| SPM1(A) | SPM1(B) | SPM1(C) | SPM2(A) | SPM2(B) | SPM2(C) | SPM(D) | Transformations | | |
|---|---|---|---|---|---|---|---|---|---|
| | | | | | | | Alice | Bob | Charlie |
| $|\Phi^+\rangle_9$ | $|\Phi^+\rangle_5$ | $|\Phi^+\rangle_1$ | $|\Psi^+\rangle_{a'}$ | $|\Psi^+\rangle_{b''}$ | $|\Psi^+\rangle_{c'''}$ | $|0\rangle_{10}$ | $I^{234}$ | $I^8$ | $I^{67}$ |
| | | | | | | | $|1\rangle_{10}$ | $\sigma_z^2 \otimes \sigma_x^{34}$ | $-i\sigma_y^8$ | $i\sigma_y^6 \otimes \sigma_x^7$ |
| $|\Phi^+\rangle_9$ | $|\Phi^+\rangle_5$ | $|\Phi^+\rangle_1$ | $|\Psi^-\rangle_{a'}$ | $|\Psi^-\rangle_{b''}$ | $|\Psi^-\rangle_{c'''}$ | $|0\rangle_{10}$ | $\sigma_z^{234}$ | $\sigma_z^8$ | $\sigma_z^{67}$ |
| | | | | | | | $|1\rangle_{10}$ | $I^2 \otimes \sigma_x^{34}$ | $-\sigma_x^8$ | $\sigma_x^{67}$ |
| $|\Phi^-\rangle_9$ | $|\Phi^-\rangle_5$ | $|\Phi^-\rangle_1$ | $|\Psi^+\rangle_{a'}$ | $|\Psi^+\rangle_{b''}$ | $|\Psi^+\rangle_{c'''}$ | $|0\rangle_{10}$ | $-i\sigma_y^2 \otimes \sigma_x^{34}$ | $-i\sigma_y^8$ | $-i\sigma_y^6 \otimes \sigma_x^7$ |
| | | | | | | | $|1\rangle_{10}$ | $\sigma_x^2 \otimes I^{34}$ | $-I^8$ | $I^{67}$ |
| $|\Phi^-\rangle_9$ | $|\Phi^-\rangle_5$ | $|\Phi^-\rangle_1$ | $|\Psi^-\rangle_{a'}$ | $|\Psi^-\rangle_{b''}$ | $|\Psi^-\rangle_{c'''}$ | $|0\rangle_{10}$ | $\sigma_x^{234}$ | $\sigma_x^8$ | $\sigma_x^{67}$ |
| | | | | | | | $|1\rangle_{10}$ | $\sigma_x^2\sigma_z^2 \otimes I^{34}$ | $\sigma_z^8$ | $\sigma_z^{67}$. |

### 2.3. Experimental Realization in IBM QE

The implementation of a particular protocol is made solely in view of the number of qubits the device supports. It provides free access through a cloud-based web-interface called IBM Quantum Experience (IBM QE), which allows researchers to design, test and run their experiments. Based on the high level of control that can be reached on the IBM digital quantum computer, we perform a proof-of-principle experimental realization of the four-party controlled asymmetric RSP scheme in IBM quantum computer. Randomness intrinsic in quantum mechanics is shown with the help of measuring an equal super position state in the computational basis for single run. The experiment is repeated for a different number of shots in the device to get the probability of obtaining each output state accurately.

The single-qubit quantum gates like Identity gate(I), Pauli gates (X, Y, Z), Hardman gate (H), and phase gates (S, $S^\dagger$, T, $T^\dagger$) are available in the IBM quantum experience toolbox. These gates can be inserted anywhere in the circuit utilizing a graphical user interface which allows click, drag, and drop method.

In the Four-party controlled cyclic asymmetrical RSP protocol of sequentially increasing qubits states, we assume that the measurement basis are $|\Phi^+\rangle_9, |\Phi^+\rangle_5, |\Phi^+\rangle_1, |\Psi^+\rangle_{a'}$, $|\Psi^+\rangle_{b''}, |\Psi^+\rangle_{c'''}$, and the coefficient of measurement basis is $a_0, a_1, b_0, b_1, c_0, c_1 = \frac{1}{\sqrt{2}}$. Owing to the q-sphere simulation is only available for circuits using less than 6 qubits, we suppose that the phase coefficient of measurement basis is $\theta_0, \theta_1, \phi_0, \phi_1, \gamma_0, \gamma_1 = 0$. We implement controlled Z gate on qubits 2, 6, and 8, and controlled X gate on qubits 3, 4, 6, 7, 8 to achieve the above scheme. The circuit diagram constructed is shown in Figure 4:

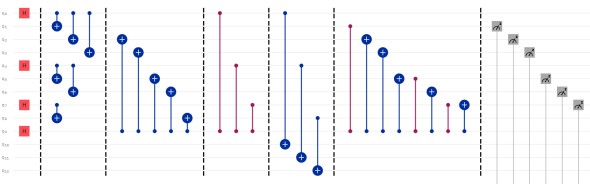

**Figure 4.** Quantum circuit illustrating four-party controlled cyclic asymmetrical RSP protocol.

The output state of the four-party controlled cyclic asymmetrical RSP protocol can be obtained by performing a measurement operator at the end of the lines, which in turn gives the average probability of each output state.

From Figure 5, it shows the state of qubits 1,5,9,10,11,12,13 are $|0\rangle$. In the ideal situation, the mean probabilities for the output state of qubits 2, 3, 4, 6, 7, 8 are 0.125, the output state is $|\varphi\rangle_{234678} = \frac{1}{2\sqrt{2}}(|000\rangle + |111\rangle) \otimes (|00\rangle + |11\rangle) \otimes (|0\rangle+|1\rangle)$, and the result after running ten times in Jupyter Notebook (Figure 5b) is more stable than the result after running once

in the ibmq_qsam_simulator (Figure 5a). We successfully implement quantum circuit in the IBM quantum computer to realize the cyclic asymmetric RSP scheme.

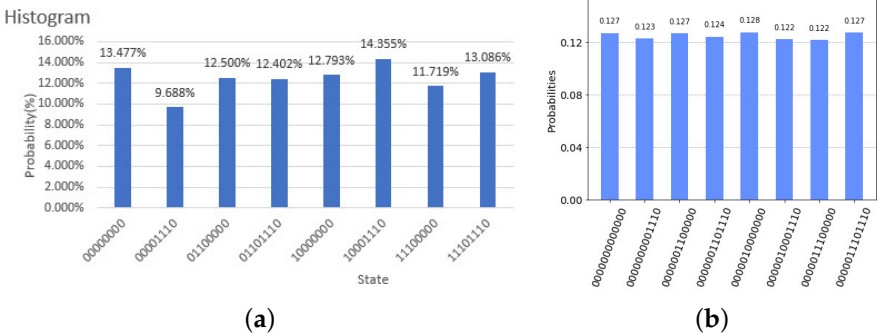

(**a**)

(**b**)

**Figure 5.** Histogram of output results. (**a**) Histogram of output results after running once in the 'ibmq_qasm_simulator' quantum processor. The x-axis of this histogram is arranged in the order of $(8, \cdots 2, 1)$ from the bottom to the top. (**b**) Histogram of output results after running ten times in Jupyter Notebook. The x-axis of this histogram is arranged in the order of $(c''', b'', \cdots 2, 1)$ from the bottom to the top.

In Table 2, we have shown the outcome of 10 runs for the output state each run having 8192 shots. According the average value and standard deviation, we obtain the average error between the ideal results and the experimental results is 0.004. The discrepancy arises from the finite coherence time of the qubits, the implementation of the gates and the process of reading out the qubits.

**Table 2.** Data analysis of the output state, where A is short for the average value, S is for standard deviation.

| Runs | $\lvert000000\rangle_{234678}$ | $\lvert000001\rangle_{234678}$ | $\lvert000110\rangle_{234678}$ | $\lvert000111\rangle_{234678}$ | $\lvert111000\rangle_{234678}$ | $\lvert111001\rangle_{234678}$ | $\lvert111110\rangle_{234678}$ | $\lvert111111\rangle_{234678}$ |
|---|---|---|---|---|---|---|---|---|
| 1 | 0.125 | 0.117 | 0.126 | 0.118 | 0.118 | 0.138 | 0.130 | 0.128 |
| 2 | 0.132 | 0.125 | 0.122 | 0.121 | 0.129 | 0.131 | 0.119 | 0.121 |
| 3 | 0.120 | 0.128 | 0.124 | 0.127 | 0.129 | 0.118 | 0.126 | 0.127 |
| 4 | 0.125 | 0.123 | 0.130 | 0.127 | 0.123 | 0.127 | 0.120 | 0,125 |
| 5 | 0.132 | 0.123 | 0.129 | 0.122 | 0.124 | 0.118 | 0.125 | 0.127 |
| 6 | 0.128 | 0.129 | 0.122 | 0.125 | 0.124 | 0.122 | 0.129 | 0.121 |
| 7 | 0.124 | 0.126 | 0.124 | 0.129 | 0.122 | 0.119 | 0.132 | 0.123 |
| 8 | 0.121 | 0.121 | 0.123 | 0.130 | 0.127 | 0.126 | 0.124 | 0.127 |
| 9 | 0.120 | 0.131 | 0.122 | 0.129 | 0.121 | 0.125 | 0.120 | 0.132 |
| 10 | 0.123 | 0.130 | 0.126 | 0.125 | 0.125 | 0.124 | 0.124 | 0.124 |
| A | 0.125 | 0.1253 | 0.1248 | 0.1253 | 0.1242 | 0.1248 | 0.1249 | 0.1255 |
| S | 0.0044 | 0.0044 | 0.0029 | 0.0039 | 0.0035 | 0.0063 | 0.0045 | 0.0039 |

## 3. Multi-Party Controlled Cyclic Asymmetrical RSP Protocol of Sequentially Increasing Qubits States

The controlled cyclic asymmetrical RSP protocol of sequentially increasing qubits states can be extended to multiparty scenario easily. Suppose there are n players $Alice_1$, $Alice_2$, ... and $Alice_n$, and controller Charlie. $Alice_i$ can prepare an i-qubit state to his neighbor $Alice_{i+1}$ under the control of Charlie. The entangled channel share among participants is $\left\lvert \varphi(\sum\limits_{i=1}^{n}(i+1)+1) \right\rangle$.

$$\left| \varphi \left( \sum_{i=1}^{n} (i+1) + 1 \right) \right\rangle$$
$$= \frac{1}{\sqrt{2}} [ (|0\cdots0\rangle + |1\cdots1\rangle)_{A_n^1, A_1^1, \cdots, A_1^n} \otimes (|0\cdots0\rangle + |1\cdots1\rangle)_{A_{n-1}^1, A_n^2, \cdots, A_n^n} \cdots \otimes (|0\cdots0\rangle$$
$$+ |1\cdots1\rangle)_{A_i^1, A_{i+1}^2, \cdots A_{i+1}^{i+1}} \cdots \otimes (|000\rangle + |111\rangle)_{A_2^1, A_3^2, A_3^3} \otimes (|00\rangle + |11\rangle)_{A_1^{n+1}, A_2^2} \otimes |0\rangle_{(\sum_{i=1}^{n} (i+1)+1)}$$
$$+ (|0\cdots1\rangle - |1\cdots0\rangle)_{A_n^1, A_1^1, \cdots, A_1^n} \otimes (|0\cdots1\rangle - |1\cdots0\rangle)_{A_{n-1}^1, A_n^2, \cdots, A_n^n} \cdots \otimes (|0\cdots1\rangle$$
$$- |1\cdots0\rangle)_{A_i^1, A_{i+1}^2, \cdots A_{i+1}^{i+1}} \cdots \otimes (|011\rangle - |100\rangle)_{A_2^1, A_3^2, A_3^3} \otimes (|01\rangle - |10\rangle)_{A_1^{n+1}, A_2^2} \otimes |1\rangle_{(\sum_{i=1}^{n} (i+1)+1)} ], \tag{21}$$

where $A_1^n$ represents the nth qubit owned by *Alice*$_1$, $A_n^n$ represents the nth qubit owned by *Alice*$_n$, and $A_{i+1}^i$ represents the *i*-th qubit owned by *Alice*$_{i+1}$, and $i \in \{1, 2, \cdots, n\}$. Charlie holds control qubit labeled by $\sum_{i=1}^{n} (i+1) + 1$. For example, if $n = 3$, the entangled channel shared should have the form as Equation (1).

If we want to change the direction of the controlled cyclic asymmetrical RSP protocol, we need to change the distribution of qubits, and the quantum channel will be changed to the following expression.

$$\left| \varphi \left( \sum_{i=1}^{n} (i+1) + 1 \right) \right\rangle$$
$$= \frac{1}{\sqrt{2}} [ (|0\cdots0\rangle + |1\cdots1\rangle)_{A_2^1, A_1^1, \cdots, A_1^n} \otimes (|0\cdots0\rangle + |1\cdots1\rangle)_{A_3^1, A_2^2, \cdots A_2^n} \cdots \otimes (|0\cdots0\rangle$$
$$+ |1\cdots1\rangle)_{A_{i+1}^1, A_i^2, \cdots A_i^{i+1}} \cdots \otimes (|000\rangle + |111\rangle)_{A_n^1, A_{n-1}^2, A_{n-1}^3} \otimes (|00\rangle + |11\rangle)_{A_1^{n+1}, A_n^2}$$
$$\otimes |0\rangle_{(\sum_{i=1}^{n} (i+1)+1)} + (|0\cdots1\rangle - |1\cdots0\rangle)_{A_2^1, A_1^1, \cdots, A_1^n} \otimes (|0\cdots1\rangle - |1\cdots0\rangle)_{A_3^1, A_2^2, \cdots A_2^n}$$
$$\cdots \otimes (|0\cdots1\rangle - |1\cdots0\rangle)_{A_{i+1}^1, A_i^2, \cdots A_i^{i+1}} \cdots \otimes (|011\rangle - |100\rangle)_{A_n^1, A_{n-1}^2, A_{n-1}^3} \otimes (|01\rangle$$
$$- |10\rangle)_{A_1^{n+1}, A_n^2} \otimes |1\rangle_{(\sum_{i=1}^{n} (i+1)+1)} ], \tag{22}$$

where $A_2^1$ represents the 1th qubit owned by *Alice*$_2$, $A_i^{i+1}$ represents the $(i + 1)$th qubit owned by *Alice*$_i$, Charlie holds control qubit labeled by $\sum_{i=1}^{n} (i+1) + 1$. Specifically, there is $(\left| \underbrace{0\cdots0}_{n+1} \right\rangle + \left| \underbrace{1\cdots1}_{n+1} \right\rangle)_{A_2^1, A_1^1, \cdots, A_1^n}$. When n is even, there is $(\left| \underbrace{0\cdots}_{\left[\frac{n+1}{2}\right]} \underbrace{\cdots1}_{\left[\frac{n+1}{2}\right]+1} \right\rangle -$

$\left| \underbrace{1\cdots}_{\left[\frac{n+1}{2}\right]} \underbrace{\cdots0}_{\left[\frac{n+1}{2}\right]+1} \right\rangle)_{A_2^1, A_1^1, \cdots, A_1^n}$. When n is odd, there is $(\left| \underbrace{0\cdots}_{\frac{n+1}{2}} \underbrace{\cdots1}_{\frac{n+1}{2}} \right\rangle - \left| \underbrace{1\cdots}_{\frac{n+1}{2}} \underbrace{\cdots0}_{\frac{n+1}{2}} \right\rangle)_{A_2^1, A_1^1, \cdots, A_1^n}$.

Available online: (accessed on 15 March 2012)

## 4. Four-Party Controlled Cyclic Asymmetrical RSP Protocol in Noisy Environments

In this section, we discuss our scheme in four noisy environments, including amplitude-damping, phase-damping noise, bit-flip noise, and phase-flip noise.

The content shared by the participants in advance is constructed by an organization called the QDC (Qubits Distribution Center). After preparing the quantum channel, QDC allocates the qubits $A_1, A_2, A_3, A_4$ to Alice, the qubits $B_1, B_2$ to Bob, and sends the qubits $C_1, C_2, C_3$ to Charlie, sends the qubits $D_1$ to David. If the quantum channel is in an ideal environment without noise influence, it is a pure state, and its density matrix can be expressed as $\rho = |\zeta\rangle\langle\zeta|$. In practical applications, when affected by a noisy environment [38],

the pure state may be converted to a mixed state. The corresponding density matrix can be written as follows:

$$
\begin{aligned}
\Lambda^q(\rho) = \sum_m & \left(E_j^q\right)_{C_1}\left(E_j^q\right)_{A_1}\left(E_j^q\right)_{A_2}\left(E_j^q\right)_{A_3}\left(E_j^q\right)_{B_1}\left(E_j^q\right)_{C_2}\left(E_j^q\right)_{C_3}\left(E_j^q\right)_{B_2} \\
& \times \left(E_j^q\right)_{A_4}\left(E_j^q\right)_{D_1}\rho\left(E_j^q\right)^\dagger_{C_1}\left(E_j^q\right)^\dagger_{A_1}\left(E_j^q\right)^\dagger_{A_2}\left(E_j^q\right)^\dagger_{A_3}\left(E_j^q\right)^\dagger_{B_1}\left(E_j^q\right)^\dagger_{C_2} \\
& \times \left(E_j^q\right)^\dagger_{C_3}\left(E_j^q\right)^\dagger_{B_2}\left(E_j^q\right)^\dagger_{A_4}\left(E_j^q\right)^\dagger_{D_1},
\end{aligned}
\tag{23}
$$

where $j$ is a label, and different noise channels have different values; the superscript indicates that the operator E acts on different noise environment and $q \in \{D, A, P, W, B, S\}$. If $q = D$, for depolarized noise, then $j = 0, 1, 2, 3$; if $q = A$, for amplitude damping noise, then $j = 0, 1$; if $q = P$, for phase damping noise, then $j = 0, 1$; if $q = S$, for bit-phase flip noise, then $j = 0, 1$.

For reconstructing the desired state, three participants and the controller have to perform single-qubit measurements based on appropriate measurement bases. The desired states were then reconstructed via unitary transformations. If the seven measurement bases are $|\Phi^+\rangle_9$, $|\Psi^+\rangle_{a'}$, $|\Phi^+\rangle_5$, $|\Psi^+\rangle_{b''}$, $|\Phi^+\rangle_1$, $|\Psi^+\rangle_{c'''}$, $|0\rangle_{10}$, then the corresponding unitary transformations would be $I_{A_2} \otimes I_{A_3} \otimes I_{A_4} \otimes I_{B_2} \otimes I_{B_3} \otimes I_{C_2}$ to recover the desired states. We obtain the reconstructed desired state in an ideal environment as follows,

$$
|\zeta\rangle = \begin{cases}
\left[c_0 e^{i\gamma_0}|000\rangle + c_1 e^{i\gamma_1}|111\rangle\right]_{A_1 A_2 A_3} \otimes \left[b_0 e^{i\phi_0}|00\rangle + b_1 e^{i\phi_1}|11\rangle\right]_{C_2 C_3} \\
\otimes \left[a_0 e^{i\theta_0}|0\rangle + a_1 e^{i\theta_1}|1\rangle\right]_{B_2}\left[|0\rangle\right]_{D_1} \\
+ \left[c_0 e^{i\gamma_0}|011\rangle - c_1 e^{i\gamma_1}|100\rangle\right]_{A_1 A_2 A_3} \otimes \left[b_0 e^{i\phi_0}|11\rangle - b_1 e^{i\phi_1}|00\rangle\right]_{C_2 C_3} \\
\otimes \left[a_1 e^{i\theta_1}|0\rangle - a_0 e^{i\theta_0}|1\rangle\right]_{B_2}\left[|1\rangle\right]_{D_1}.
\end{cases}
\tag{24}
$$

For any two quantum states, including pure state and mixed state, the fidelity is $F(\rho, \sigma) = Tr\sqrt{\rho^{\frac{1}{2}}\sigma\rho^{\frac{1}{2}}}$, which is defined as a formula composed of density operators in two states. Then the fidelity between two pure states $|\psi\rangle$ and $|\varphi\rangle$ is the modulus of their inner product, which is $F = |\langle\psi \mid \varphi\rangle|$. As in this article, when the ideal output state is a pure state and the observed output state is a mixed state, the fidelity formula defined by the original fidelity is simplified to $F = \langle\zeta|\rho_{out}|\zeta\rangle$. When the two states are mixed state, the density matrix can be calculated separately employing the universal formula $F(\rho, \sigma) = Tr(\rho\sigma) + \sqrt{1 - Tr(\rho^2)}\sqrt{1 - Tr(\sigma^2)}$.

### 4.1. Depolarized Noise

For a single qubit, supposing that the qubit is depolarized with probability p and remains unchanged with probability 1-p. The effect of the depolarization noise on the Bloch sphere is the entire sphere shrinks uniformly as a function of p. Then the state of the quantum system in depolarized noise is $\varepsilon(\rho) = \frac{pI}{2} + (1-p)\rho$, and supposing there is $\frac{I}{2} = \frac{\rho + X\rho X + Y\rho Y + Z\rho Z}{4}$, and then get $\varepsilon(\rho) = (1-p)\rho + \frac{p}{3}(X\rho X + Y\rho Y + Z\rho Z)$. The depolarized noise can be expressed by Kraus operators given in Formula (25).

$$
E_0^D = \sqrt{1 - \frac{3}{4}p_D}, \quad E_1^D = \sqrt{\frac{p_D}{4}}\begin{bmatrix} 0 & 1 \\ 1 & 0 \end{bmatrix}, \quad E_2^D = \sqrt{\frac{p_D}{4}}\begin{bmatrix} 0 & -i \\ i & 0 \end{bmatrix}, \quad E_3^D = \sqrt{\frac{p_D}{4}}\begin{bmatrix} 1 & 0 \\ 0 & -1, \end{bmatrix}
\tag{25}
$$

where $p_D \in [0, 1]$ is the decoherence rate for depolarized noise. $E_0^D$ means that the quantum state $|0\rangle$, $|1\rangle$ is not changed, but the amplitude of the quantum state is reduced; $E_1^D$ means the quantum state $|0\rangle$ and $|1\rangle$ transition with probability $\sqrt{\frac{p_D}{4}}$; $E_2^D$ means the quantum state $|0\rangle$ transitions to $i|1\rangle$, the quantum state $|1\rangle$ transitions to $-i|0\rangle$; $E_2^D$ means the quantum state $|1\rangle$ transition to $|-1\rangle$.

If the scheme is implemented in depolarized noisy environment, then the concrete density matrices of the output states can be obtained as follows.

$$(\rho_{out}^D) = \begin{cases}
(1-\frac{3}{4}p_D)^{10}\big\{ \big[c_0e^{i\gamma_0}|000\rangle + c_1e^{i\gamma_1}|111\rangle\big]_{A_1A_2A_3} \times \big[b_0e^{i\phi_0}|00\rangle + b_1e^{i\phi_1}|11\rangle\big]_{C_2C_3} \\
\times \big[a_0e^{i\theta_0}|0\rangle + a_1e^{i\theta_1}|1\rangle\big]_{B_2}[|0\rangle]_{D_1} + \big[c_0e^{i\gamma_0}|011\rangle - c_1e^{i\gamma_1}|100\rangle\big]_{A_1A_2A_3} \\
\times \big[b_0e^{i\phi_0}|11\rangle - b_1e^{i\phi_1}|00\rangle\big]_{C_2C_3} \times \big[a_1e^{i\theta_1}|0\rangle - a_0e^{i\theta_0}|1\rangle\big]_{B_2}[|1\rangle]_{D_1}\big\} \\
\times \big[c_0e^{-i\gamma_0}\langle000| + c_1e^{-i\gamma_1}\langle111|\big]_{A_1A_2A_3} \times \big[b_0e^{-i\phi_0}\langle00| + b_1e^{-i\phi_1}\langle11|\big]_{C_2C_3} \\
\times \big[a_0e^{-i\theta_0}\langle0| + a_1e^{-i\theta_1}\langle11|\big]_{B_2}[\langle0|]_{D_1} + \big[c_0e^{-i\gamma_0}\langle011| - c_1e^{-i\gamma_1}\langle100|\big]_{A_1A_2A_3} \\
\times \big[b_0e^{-i\phi_0}\langle11| - b_1e^{-i\phi_1}\langle00|\big]_{C_2C_3} \times \big[a_1e^{-i\theta_1}\langle0| - a_0e^{-i\theta_0}\langle1|\big]_{B_2}[\langle1|]_{D_1}\big\} \\
+(\frac{p_D}{4})^{10}\big\{ \big[c_1e^{i\gamma_1}|111\rangle + c_0e^{i\gamma_0}|000\rangle\big]_{A_1A_2A_3} \times \big[b_1e^{i\phi_1}|11\rangle + b_0e^{i\phi_0}|00\rangle\big]_{C_2C_3} \\
\times \big[a_1e^{i\theta_1}|1\rangle + a_0e^{i\theta_0}|0\rangle\big]_{B_2}[|1\rangle]_{D_1} + \big[c_1e^{i\gamma_1}|100\rangle - c_0e^{i\gamma_0}|011\rangle\big]_{A_1A_2A_3} \\
\times \big[b_1e^{i\phi_1}|00\rangle - b_0e^{i\phi_0}|11\rangle\big]_{C_2C_3} \times \big[a_0e^{i\theta_0}|1\rangle - a_1e^{i\theta_1}|0\rangle\big]_{B_2}[|0\rangle]_{D_1} \\
\times \big[c_1e^{-i\gamma_1}\langle111| + c_0e^{-i\gamma_0}\langle000|\big]_{A_1A_2A_3} \times \big[b_1e^{-i\phi_1}\langle11| + b_0e^{-i\phi_0}\langle00|\big]_{C_2C_3} \\
\times \big[a_1e^{-i\theta_1}\langle1| + a_0e^{-i\theta_0}\langle0|\big]_{B_2}[\langle1|]_{D_1} + \big[c_1e^{-i\gamma_1}\langle100| - c_0e^{-i\gamma_0}\langle011|\big]_{A_1A_2A_3} \\
\times \big[b_1e^{-i\phi_1}\langle00| - b_0e^{-i\phi_0}\langle11|\big]_{C_2C_3} \times \big[a_0e^{-i\theta_0}\langle1| - a_1e^{-i\theta_1}\langle0|\big]_{B_2}[\langle0|]_{D_1}\big\} \\
+(\frac{p_D}{4})^{10}\big\{ \big[c_1e^{i\gamma_1}|111\rangle + c_0e^{i\gamma_0}|000\rangle\big]_{A_1A_2A_3} \times \big[-i(b_1e^{i\phi_1}|11\rangle - b_0e^{i\phi_0}|00\rangle)\big]_{C_2C_3} \\
\times \big[-(a_1e^{i\theta_1}|1\rangle + a_0e^{i\theta_0}|0\rangle)\big]_{B_2}[i|1\rangle]_{D_1} + \big[c_1e^{i\gamma_1}|100\rangle - c_0e^{i\gamma_0}|011\rangle\big]_{A_1A_2A_3} \\
\times \big[-i(b_1e^{i\phi_1}|00\rangle + b_0e^{i\phi_0}|11\rangle)\big]_{C_2C_3} \times \big[a_0e^{i\theta_0}|1\rangle - a_1e^{i\theta_1}|0\rangle\big]_{B_2}[-i|0\rangle]_{D_1} \\
\times \big[c_1e^{-i\gamma_1}\langle1111| + c_0e^{-i\gamma_0}\langle000|\big]_{A_1A_2A_3} \times \big[-i(b_1e^{-i\phi_1}\langle11| - b_0e^{-i\phi_0}\langle00|)\big]_{C_2C_3} \\
\times \big[-(a_1e^{-i\theta_1}\langle1| + a_0e^{-i\theta_0}\langle0|)\big]_{B_2}[i\langle1|]_{D_1} + \big[c_1e^{-i\gamma_1}\langle100| - c_0e^{-i\gamma_0}\langle011|\big]_{A_1A_2A_3} \\
\times \big[-i(b_1e^{-i\phi_1}\langle00| + b_0e^{-i\phi_0}\langle11|)\big]_{C_2C_3} \times \big[a_0e^{-i\theta_0}\langle1| - a_1e^{-i\theta_1}\langle0|)\big]_{B_2}[-i\langle0|]_{D_1}\big\} \\
+(\frac{p_D}{4})^{10}\big\{ \big[c_0e^{i\gamma_0}|000\rangle + c_1e^{i\gamma_1}|111\rangle\big]_{A_1A_2A_3} \times \big[b_0e^{i\phi_0}|00\rangle - b_1e^{i\phi_1}|11\rangle\big]_{C_2C_3} \\
\times \big[a_0e^{i\theta_0}|0\rangle + a_1e^{i\theta_1}|1\rangle\big]_{B_2}[|0\rangle]_{D_1} + \big[c_0e^{i\gamma_0}|011\rangle - c_1e^{i\gamma_1}|100\rangle\big]_{A_1A_2A_3} \\
\times \big[b_0e^{i\phi_0}|11\rangle + b_1e^{i\phi_1}|00\rangle\big]_{C_2C_3} \times \big[-(a_1e^{i\theta_1}|0\rangle - a_0e^{i\theta_0}|1\rangle)\big]_{B_2}[-|1\rangle]_{D_1} \\
\times \big[c_0e^{-i\gamma_0}\langle000| + c_1e^{-i\gamma_1}\langle111|\big]_{A_1A_2A_3} \times \big[b_0e^{-i\phi_0}\langle00| - b_1e^{-i\phi_1}\langle11|\big]_{C_2C_3} \\
\times \big[a_0e^{-i\theta_0}\langle0| + a_1e^{-i\theta_1}\langle1|\big]_{B_2}[\langle0|]_{D_1} + \big[c_0e^{-i\gamma_0}\langle011| - c_1e^{-i\gamma_1}\langle100|\big]_{A_1A_2A_3} \\
\times \big[b_0e^{-i\phi_0}\langle11| + b_1e^{-i\phi_1}\langle00|\big]_{C_2C_3} \times \big[-(a_1e^{-i\theta_1}\langle0| - a_0e^{-i\theta_0}\langle1|)\big]_{B_2}[-\langle1|]_{D_1}\big\}.
\end{cases} \tag{26}$$

Then, the fidelity of the output state in depolarized noisy environment can be calculated as:

$$\begin{aligned}
F^D &= \langle\zeta|\rho_{out}^D|\zeta\rangle \\
&= \Big\{ (1-\tfrac{3}{4}p_D)^{10} + (\tfrac{p_D}{4})^{10}\big[ (b_0^2 - b_1^2)^2 + (b_0^2 - b_1^2)^2 \times (-a_0^2 - a_1^2)^2 \big] \Big\}.
\end{aligned} \tag{27}$$

It can be seen from the above formula that the fidelity of the output quantum state is related to the decoherence rate in depolarization noise and the coefficient of the desired quantum state.

### 4.2. Amplitude Damping Noise

The amplitude damping noise can be expressed by Kraus operators given in Formula (28).

$$E_0^A = \begin{bmatrix} 1 & 0 \\ 0 & \sqrt{1-p_A} \end{bmatrix}, E_1^A = \begin{bmatrix} 0 & \sqrt{p_A} \\ 0 & 0 \end{bmatrix}, \tag{28}$$

where $P_A = \sin^2\theta (P_A \in [0,1])$ is the decoherence rates for amplitude damping noise. $E_0^A$ means that quantum state $|0\rangle$ is not changed, and only reduce the amplitude of quantum state $|1\rangle$; $E_1^A$ means the quantum state $|1\rangle$ transition to $|0\rangle$.

If the scheme is implemented in amplitude damping noisy environment, then the concrete density matrices of the output states can be obtained as follows.

$$
(\rho_{out}^A) = \left\{
\begin{aligned}
&\{\left[c_0 e^{i\gamma_0}|000\rangle + (1-P_A)^2 c_1 e^{i\gamma_1}|111\rangle\right]_{A_1 A_2 A_3} \\
&\times \left[b_0 e^{i\phi_0}|00\rangle + (\sqrt{1-P_A})^3 b_1 e^{i\phi_1}|11\rangle\right]_{C_2 C_3} \\
&\times \left[a_0 e^{i\theta_0}|0\rangle + (1-P_A)a_1 e^{i\theta_1}|1\rangle\right]_{B_2}[|0\rangle]_{D_1} \\
&+ \left[(1-P_A)(c_0 e^{i\gamma_0}|011\rangle - c_1 e^{i\gamma_1}|100\rangle)\right]_{A_1 A_2 A_3} \\
&\times \left[((1-P_A)b_0 e^{i\phi_0}|11\rangle - \sqrt{1-P_A}b_1 e^{i\phi_1}|00\rangle)\right]_{C_2 C_3} \\
&\times \left[\sqrt{1-P_A}(a_1 e^{i\theta_1}|0\rangle - a_0 e^{i\theta_0}|1\rangle)\right]_{B_2}\left[\sqrt{1-P_A}|1\rangle\right]_{D_1}\} \\
&\times \{\left[c_0 e^{-i\gamma_0}\langle 000| + (1-P_A)^2 c_1 e^{-i\gamma_1}\langle 111|\right]_{A_1 A_2 A_3} \\
&\times \left[b_0 e^{-i\phi_0}\langle 00| + (\sqrt{1-P_A})^3 b_1 e^{-i\phi_1}\langle 11|\right]_{C_2 C_3} \\
&\times \left[a_0 e^{-i\theta_0}\langle 0| + (1-P_A)a_1 e^{-i\theta_1}\langle 1|\right]_{B_2}[\langle 0|]_{D_1} \\
&+ \left[(1-P_A)(c_0 e^{-i\gamma_0}\langle 011| - c_1 e^{-i\gamma_1}\langle 100|)\right]_{A_1 A_2 A_3} \\
&\times \left[(1-P_A)(b_0 e^{-i\phi_0}\langle 11| - b_1 e^{-i\phi_1}\langle 00|)\right]_{C_2 C_3} \\
&\times \left[\sqrt{1-P_A}(a_1 e^{-i\theta_1}\langle 0| - a_0 e^{-i\theta_0}\langle 1|)\right]_{B_2}\left[\sqrt{1-P_A}\langle 1|\right]_{D_1}\} \\
&+ P_A{}^{10} c_0^2 b_0^2 a_0^2 (|0000000\rangle + |0000000\rangle)_{A_1 A_2 A_3 C_2 C_3 B_2 D_1} \\
&\times (\langle 0000000| + \langle 0000000|)_{A_1 A_2 A_3 C_2 C_3 B_2 D_1}.
\end{aligned}
\right.
\tag{29}
$$

Then, the fidelity of the output state in amplitude damping noisy environment can be calculated as:

$$
\begin{aligned}
F^A &= \langle \zeta | \rho_{out}^A | \zeta \rangle \\
&= \left\{
\begin{aligned}
&\tfrac{1}{2}\{\{\left[c_0^2 + (1-P_A)^2 c_1^2\right] \times \left[b_0^2 + (\sqrt{1-P_A})^3 b_1^2\right] \times \left[a_0^2 + (1-P_A)a_1^2\right]\}^2 \\
&+ \{\left[(1-P_A)(c_0^2 + c_1^2)\right] \times \left[((1-P_A)b_0^2 + \sqrt{1-P_A}b_1^2)\right] \\
&\times \left[\sqrt{1-P_A}(a_0^2 + a_1^2)\right] \times (1-P_A)\}^2\} + P_A^{10} c_0^4 b_0^4 a_0^4.
\end{aligned}
\right\}
\end{aligned}
\tag{30}
$$

It can be seen from the above formula that the fidelity of the output quantum state is related to the decoherence rate in amplitude damping noise and the coefficient of the desired quantum state.

### 4.3. Phase Damping Noise

The phase damping noise can be expressed by Kraus operators given in Formula (31).

$$
E_0^P = \sqrt{1-p_P}\begin{bmatrix}1 & 0 \\ 0 & 1\end{bmatrix}, E_1^P = \sqrt{p_P}\begin{bmatrix}1 & 0 \\ 0 & 0\end{bmatrix}, E_2^P = \sqrt{p_P}\begin{bmatrix}0 & 0 \\ 0 & 1\end{bmatrix},
\tag{31}
$$

where $p_P \in [0,1]$ is the decoherence rates for phase damping noise.

If the scheme is implemented in phase damping noisy environment, then the concrete density matrices of the output states can be obtained as follows.

$$
(\rho_{out}^P) = \left\{
\begin{aligned}
&(1-p_P)^{10}\{\left[c_0 e^{i\gamma_0}|000\rangle + c_1 e^{i\gamma_1}|111\rangle\right]_{A_1 A_2 A_3} \times \left[b_0 e^{i\phi_0}|00\rangle + b_1 e^{i\phi_1}|11\rangle\right]_{C_2 C_3} \\
&\times \left[a_0 e^{i\theta_0}|0\rangle + a_1 e^{i\theta_1}|1\rangle\right]_{B_2}|0\rangle_{D_1} + \left[c_0 e^{i\gamma_0}|011\rangle - c_1 e^{i\gamma_1}|100\rangle)\right]_{A_1 A_2 A_3} \\
&\times \left[b_0 e^{i\phi_0}|11\rangle - b_1 e^{i\phi_1}|00\rangle\right]_{C_2 C_3} \times \left[a_1 e^{i\theta_1}|0\rangle - a_0 e^{i\theta_0}|1\rangle\right]_{B_2}|1\rangle_{D_1}\} \\
&\times \left[c_0 e^{-i\gamma_0}\langle 000| + c_1 e^{-i\gamma_1}\langle 111|\right]_{A_1 A_2 A_3} \times \left[b_0 e^{-i\phi_0}\langle 00| + b_1 e^{-i\phi_1}\langle 11|\right]_{C_2 C_3} \\
&\times \left[a_0 e^{-i\theta_0}\langle 0| + a_1 e^{-i\theta_1}\langle 1|\right]_{B_2}[\langle 0|]_{D_1} + \left[c_0 e^{-i\gamma_0}\langle 011| - c_1 e^{-i\gamma_1}\langle 100|\right]_{A_1 A_2 A_3} \\
&\times \left[b_0 e^{-i\phi_0}\langle 11| - b_1 e^{-i\phi_1}\langle 00|\right]_{C_2 C_3} \times \left[a_1 e^{-i\theta_1}\langle 0| - a_0 e^{-i\theta_0}\langle 1|\right]_{B_2}\langle 1|_{D_1}\} \\
&+ p_P^{10} c_0^2 b_0^2 a_0^2 [|0000000\rangle + |0000000\rangle]_{A_1 A_2 A_3 C_2 C_3 B_2 D_1} \\
&\times [\langle 0000000| + \langle 0000000|]_{A_1 A_2 A_3 C_2 C_3 B_2 D_1} \\
&+ p_P^{10} c_1^2 b_1^2 a_1^2 [|1111111\rangle + |1111111\rangle]_{A_1 A_2 A_3 C_2 C_3 B_2 D_1} \\
&\times [\langle 1111111| + \langle 1111111|]_{A_1 A_2 A_3 C_2 C_3 B_2 D_1}.
\end{aligned}
\right.
\tag{32}
$$

Then, the fidelity of the output state in phase damping noisy environment can be calculated as:

$$F^P = \langle \zeta | \rho_{out}^P | \zeta \rangle = (1 - p_P)^{10} + p_P^{10} c_0^4 b_0^4 a_0^4 + p_P^{10} c_1^4 b_1^4 a_1^4. \tag{33}$$

It can be seen from the above formula that the fidelity of the output quantum state is related to the coefficient of the desired quantum state and the decoherence rate in phase damping noise.

### 4.4. Bit-Phase Flip Noise

The bit-phase flip noise is a combination of a phase flip and a bit flip, and its influence to the quantum channel can be expressed by the Kraus operators given in Formula (34).

$$E_0^S = \sqrt{1 - s_P} \begin{bmatrix} 1 & 0 \\ 0 & 1 \end{bmatrix}, E_1^S = \sqrt{s_P} \begin{bmatrix} 0 & -i \\ i & 0 \end{bmatrix}, \tag{34}$$

where $p_S \in [0, 1]$ is the decoherence rates for bit-phase flip noise. $E_0^S$ means that the quantum state $|0\rangle$ is not changed, but the amplitude of quantum state $|1\rangle$ is reduced; $E_1^S$ means that quantum state $|0\rangle$ transitions to $i|1\rangle$, quantum state $|1\rangle$ transitions to $-i|0\rangle$.

If the scheme is implemented in bit-phase flip noisy environment, then the concrete density matrices of the output states can be obtained as follows.

$$(\rho_{out}^S) = \left\{ \begin{array}{l} (1 - p_S)^{10} \{ [c_0 e^{i\gamma_0} |000\rangle + c_1 e^{i\gamma_1} |111\rangle]_{A_1 A_2 A_3} \times [b_0 e^{i\phi_0} |00\rangle + b_1 e^{i\phi_1} |11\rangle]_{C_2 C_3} \\ \times [a_0 e^{i\theta_0} |0\rangle + a_1 e^{i\theta_1} |1\rangle]_{B_2} |0\rangle_{D_1} + [c_0 e^{i\gamma_0} |011\rangle - c_1 e^{i\gamma_1} |100\rangle)]_{A_1 A_2 A_3} \\ \times [b_0 e^{i\phi_0} |11\rangle - b_1 e^{i\phi_1} |00\rangle]_{C_2 C_3} \times [a_1 e^{i\theta_1} |0\rangle - a_0 e^{i\theta_0} |1\rangle]_{B_2} |1\rangle_{D_1} \} \\ \times [c_0 e^{-i\gamma_0} \langle 000| + c_1 e^{-i\gamma_1} \langle 111|]_{A_1 A_2 A_3} \times [b_0 e^{-i\phi_0} \langle 00| + b_1 e^{-i\phi_1} \langle 11|]_{C_2 C_3} \\ \times [a_0 e^{-i\theta_0} \langle 0| + a_1 e^{-i\theta_1} \langle 1|]_{B_2} [\langle 0|]_{D_1} + [c_0 e^{-i\gamma_0} \langle 011| - c_1 e^{-i\gamma_1} \langle 100|]_{A_1 A_2 A_3} \\ \times [b_0 e^{-i\phi_0} \langle 11| - b_1 e^{-i\phi_1} \langle 00|]_{C_2 C_3} \times [a_1 e^{-i\theta_1} \langle 0| - a_0 e^{-i\theta_0} \langle 1|]_{B_2} \langle 1|_{D_1} \} \\ + p_S^{10} \{ [c_0 e^{i\gamma_0} |000\rangle + c_1 e^{i\gamma_1} |111\rangle]_{A_1 A_2 A_3} \times [i(b_0 e^{i\phi_0} |00\rangle - b_1 e^{i\phi_1} |11\rangle)]_{C_2 C_3} \\ \times [-a_0 e^{i\theta_0} |0\rangle - a_1 e^{i\theta_1} |1\rangle]_{B_2} [i|1\rangle]_{D_1} + [c_1 e^{i\gamma_1} |100\rangle - c_0 e^{i\gamma_0} |011\rangle)]_{A_1 A_2 A_3} \\ \times [-i b_0 e^{i\phi_0} |11\rangle - i b_1 e^{i\phi_1} |00\rangle]_{C_2 C_3} \times [a_1 e^{i\theta_1} |1\rangle - a_0 e^{i\theta_0} |0\rangle]_{B_2} [-i|0\rangle]_{D_1} \} \\ \times [c_0 e^{-i\gamma_0} \langle 000| + c_1 e^{-i\gamma_1} \langle 111|]_{A_1 A_2 A_3} \times [i(b_0 e^{-i\phi_0} \langle 00| + b_1 e^{-i\phi_1} \langle 11|)]_{C_2 C_3} \\ \times [-a_0 e^{-i\theta_0} \langle 0| - a_1 e^{-i\theta_1} \langle 1|]_{B_2} [i\langle 1|]_{D_1} + [c_1 e^{-i\gamma_1} \langle 100| - c_0 e^{-i\gamma_0} \langle 011|]_{A_1 A_2 A_3} \\ \times [-i(b_0 e^{-i\phi_0} \langle 11| - b_1 e^{-i\phi_1} \langle 00|)]_{C_2 C_3} \times [a_1 e^{-i\theta_1} \langle 1| - a_0 e^{-i\theta_0} \langle 0|]_{B_2} [-i\langle 0|]_{D_1} \}. \end{array} \right. \tag{35}$$

Then, the fidelity of the output state in bit-phase flip noisy environment can be calculated as:

$$F^S = \langle \zeta | \rho_{out}^S | \zeta \rangle = (1 - p_S)^{10}. \tag{36}$$

It can be seen from the above formula that the fidelity of the output quantum state is only related to the decoherence rate in bit-phase flip noise.

### 4.5. Analysis of the Effect of the Scheme in Four Noisy Environments

From the above calculation results of fidelity, it can be found that in the environment of depolarization, amplitude damping and phase damping noise, the fidelity of the output state depends on the coefficient of the desired state and the decoherence rate. However, in a bit-phase flip noise environment, the fidelity of the output state is only decided by the decoherence rate.

As can be seen from the Figure 6, in the depolarization noise, amplitude damped noise, and bit-phase flip noise channels, as the probability of depolarization increases, the fidelity of the output state gradually tends to 0; while in phase damping noise channel, the fidelity of the output state decreases with the increase of decoherence rate and then increases when P > 0.8.

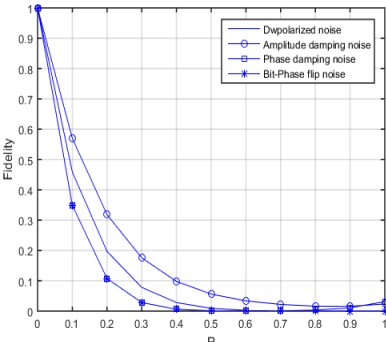

**Figure 6.** The trend of the fidelity of the output state with the change of the decoherence rate(P) in four types of noisy environments, where different types of lines represent different noise environments. Suppose the coefficient of the desired state $a_0 = a_1 = b_0 = b_1 = c_0 = c_1 = \frac{1}{\sqrt{2}}$.

When the decoherence rate of bit-phase flip noise and phase damped noise channel is 0.4, the fidelity of the output state drops to 0, and the fidelity of the phase damping noise is gradually increasing after the decoherence rate is 0.8. When the decoherence rate of the depolarized noise channel is 0.6, the fidelity of the output state drops to 0; and the amplitude damping noise is less affected by the decoherence rate than the other three noises.

When simultaneously considering the influence of the coefficient of the desired quantum state and the decoherence rate, it is assumed that the coefficients of the desired quantum state are all equal. There are the variations of fidelity in different noises as the decoherence rate and the coefficient of desired quantum state varied simultaneously is shown in the figure below:

From Figure 7, it can be observed that the Fidelity(D) and the Fidelity(S) decrease with the P(D) and P(S). The Fidelity(D) and the Fidelity(S) independent of the coefficient of the desired quantum state. In the amplitude damped noise, when the coefficient of the desired state is greater than 0.7, the Fidelity(A) is positively correlated with the transformation of P(A). In the phase damping noise channel, when the coefficient of the desired state is 1 and the P(p) is higher than 0.5, the Fidelity(p) increases as the P(p) increases. When the coefficient of the desired state is less than 1, the Fidelity(p) decreases with the increasing of P(p).

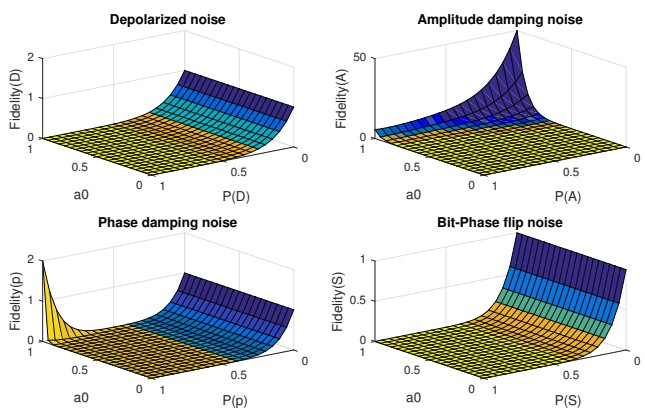

**Figure 7.** The variations of the fidelity in different noise with different coefficients of the initiated state and decoherence rate, where P(D) is short for the decoherence rate of the depolarized noise, P(A) is short for the decoherence rate of the Amplitude damping noise, P(p) is short for the decoherence rate of the phase damping noise, P(S) is short for the decoherence rate of the bit-phase flip noise. Suppose the coefficient of the desired state is $a_0 = a_1 = b_0 = b_1 = c_0 = c_1 \in [0, 1]$.

## 5. Comparison and Conclusions

Efficiency [39] is a main factor utilizing for weighing the superiority of RSP protocol. In comparison with the other cyclic RSP schemes, we calculate the scheme's efficiency ($\eta$) with the following manner.

$$\eta = \frac{q_i}{q_s + q_c},\tag{37}$$

where $q_i$ denotes the number of states to be transmitted, $q_s$ indicates the quantum resource consumption, and $q_c$ means the classical resource consumption.

From Table 3, the following conclusions can be drawn:

(1) The efficiency of our scheme is 32%, which is relatively higher than those of the schemes in [16,23–25,30,32]. Besides, our scheme has a larger transmission capacity than the other schemes because the sequentially increasing qubits could carry more information via the transmission in the scheme.

(2) Due to the influence of the real environment, the initial maximally entangled state easily evolves into a non-maximally entangled state or a mixed state. Compared with the [25], we employ a ten-qubit non-maximally entangled state as quantum channel which is more realistic than the maximally entangled state.

(3) In comparison with [25], we increase the number of states to be transmitted in the four-party cyclic remote preparation scheme of an arbitrary single-particle state, an arbitrary two-qubit state and an arbitrary three-qubit state, which is more efficient and realistic.

(4) Another remarkable advantage of our scheme is that the protocol can be employed to send multiple quantum information among multiple players either in an asymmetric manner. Compared with [16,23–25,30,32], our scheme can realize the system having N > 3 observers. When the value of N is specified, we can construct anyone of all possible quantum channels.

**Table 3.** Comparison with other RSP schemes, where BSM is short for Bell-state measurement and SM for single-qubit measurement.

| Scheme | Type | $q_i$ | $q_s$ | $q_c$ | Operation | Efficiency |
|--------|------|-------|-------|-------|-----------|------------|
| [25] | CCJRSP | 3 (three arbitrary single-qubit) | 10 | 6 | 7 SM | 19% |
| [32] | CJRSP | 3 (three arbitrary single-qubit) | 9 | 9 | 6 SM | 17% |
| [30] | CCJRSP | 2 (An arbitrary two-qubit state) | 7 | 5 | 5 SM | 17% |
| [24] | CBRSP | 4(Two arbitrary two-qubit ) | 9 | 8 | 9 SM | 24% |
| [16] | CBRSP and QT | 2 (two arbitrary single-qubit) | 7 | 4 | 1 BSM,3 SM | 18% |
| [23] | CCRSP | 3 (three arbitrary single-qubit) | 7 | 6 | 6 SM | 23% |
| Ours | CCARSP | 6(A single-qubit, a two-qubit, a three-qubit) | 13 | 6 | 7 SM | 32% |

In this paper, the most significant innovation of the proposed scheme is reflected in realizing the remote state preparation with sequentially increasing qubits among multi-party, which is not only cyclically asymmetric but also bidirectional. We introduce a specific case of our scheme concretely, namely, the four-party controlled cyclic asymmetrical RSP protocol. Then the four-party controlled cyclic asymmetrical RSP protocol is demonstrated on the IBM quantum computer by designing appropriate quantum circuits using single-qubit and two-qubit quantum gates. The desired quantum states are reconstructed by adopting appropriate unitary transformations with a success probability of 100%. Furthermore, arbitrary qubit transmission channels with different circulation directions are designed and constructed. Considering the mutual coupling between the open quantum system and the environment, the feasibility of the scheme under different noise environments is analyzed respectively.

In our subsequent research, we will consider a case where the input state is a mixed state and the output state is also a mixed state. Besides, the phase transition between the

output state and the input state in the scheme can be discussed by simulating it in IBM Quantum Experience.

**Author Contributions:** Conceptualization, N.Z. and T.W.; methodology, N.Z. and T.W.; software, T.W. and Y.Y.; validation, N.Z., T.W. and Y.Y.; formal analysis, Y.Y. and C.P.; resources, T.W.; data curation, N.Z. and T.W.; Formula derivation and calculation, T.W. and N.Z.; writing—original draft preparation, N.Z. and T.W.; writing—review and editing, N.Z., T.W. and Y.Y.; visualization, N.Z. and T.W.; supervision, C.P.; project administration, N.Z. and C.P.; funding acquisition, N.Z. and C.P. All authors have read and agreed to the published version of the manuscript.

**Funding:** This work was funded by the Shannxi Key Industrial Innovation China Project in Industrial Domain (Grant No. 2019ZDLGY09-03, No. 2020ZDLGY15-09), the National Natural Science Foundation of China (Grants No. 61771296, No. 61372076, No. 61301171), the Natural Science Foundation of Shaanxi province (Grant No. 2018JM60-53, No. 2018JZ60-06), the 111 Project under Grant B08038.

**Acknowledgments:** We wish to express their appreciation to the reviewers for their helpful suggestions which greatly improved the presentation of this paper.

**Conflicts of Interest:** The authors declares that there is no conflict of interest with this article. The funders had no role in the design of the study; in the collection, analyses, or interpretation of data; in the writing of the manuscript, or in the decision to publish the results.

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
