# Peer review of "A Scheme for Controlled Cyclic Asymmetric Remote State Preparation in Noisy Environment"

_applsci, doi:10.3390/app11041405_

Round 1
Reviewer 1 Report
In this paper, the authors consider remote state preparation (RSP), which is basically like teleportation except that the state that is to be teleported is known beforehand. This paper considers Controlled Cyclic Asymmetric Remote State Preparation in the presence of noise.
My main question is, why do the authors have to consider such a complicated way to perform RSP? In particular, why do the authors have to consider Controlled Cyclic Asymmetric Remote State Preparation when the normal RSP will do? Is there any advantage in going to this more complicated procedure?
It is nice that the authors compute the fidelities of RSP for different noise channels such as amplitude damping, phase and and bit flip errors, and depolarizing errors. However, the formulas seem to apply for pure states in RSP. Can the formulas generalize to the RSP of mixed states?
Author Response
We feel great thanks for your professional review work on our manuscript. The comments are very helpful for revising and improving our paper. They are of important guiding significance to our future scientific research. We have studied the comments carefully and made corrections.
Please see the attachment.

Reviewer 2 Report
This paper is very interesting for me.
It should be accepted.
Author Response
Response to Reviewer 2 Comments
Point 1: This paper is very interesting for me. It should be accepted.
Response 1:
Thanks for your comments.
We feel great thanks for your professional review work on our manuscript, and we are very happy to accept your positive and constructive feedbacks.
Reviewer 3 Report
The authors consider a remote state preparation scheme with four parties involved and a ten-qubit entangled state as a quantum channel. It is hard to find any significant novelty in this manuscript, since all the elements can be found in the extensive literature about this topic -only some of it cited by the authors. For instance, the protocol proposed is almost identical to International Journal of Theoretical Physics (2019) 58:255–260 with the minor addition that not all the parties prepare a single-qubit state. The analysis of the effect of noisy environments in RSP schemes can also be found in the literature and it is actually rather trivial within the abstract approach of the authors. The authors launch some experiments in the IBM quantum computer. It is not clear why RSP in a single quantum computer with non-separated parties would be interesting, unless for pedagogical or academical reasons. Moreover, the authors only show some histograms and say that the results are "close" to the ideal. Such a vague statement is not admisible in a scientific publication, some computation of the errors/fidelities should be provided. However, this is not possible with only a probability histogram, since the phases are missing. Therefore we actually do not know if the results are close -whatever this means- to the ideal, unless we assume that the phases are the same as the ideal, which we do not know. Moreover, the presentation of the results is extremely poor. Among many examples, we read for instance at the beginning of page 16 that "[...] the relationship between the output states of four noise channel and the decadence rate is:" and nothing follows. Finally, the English is unreadable at some points.
For all the reasons above, this paper cannot be published in a scientific journal.
Author Response
We feel great thanks for your professional review work on our manuscript. Those comments are very helpful for revising and improving our paper. They are of important guiding significance to our future scientific research. We have studied the comments carefully and made corrections.
Please see the attachment.

Reviewer 4 Report
The topic is interesting and the article is well written. There are a lot of formulas that cannot be verified. In fact, I see that the authors have significantly revised the article and made many changes. In this case, I see that the authors paid a lot of attention to the work. I support the authors, but I cannot say with confidence about the reliability of the conclusions in the article.
Author Response
Response to Reviewer 4 Comments
We feel great thanks for your professional review work on our manuscript. Those comments are very helpful for revising and improving our paper. They are of important guiding significance to our future scientific research.
We have studied the comments carefully and made corrections. Revised portions are marked in blue in the paper.
Point 1: The topic is interesting and the article is well written. There are a lot of formulas that cannot be verified.
Response 1: Thanks for your comments. The comments are all valuable and very helpful for revising and improving our paper.
The formulas in our paper are based on the definitions in the book “Quantum Computation and Quantum Information” edited by Michael A.Nielsen and Isaac L.Chuang.
As reviewer suggested that we perform calculations to the formulas in the paper and confirm that formulas are correct.
Point 2: I support the authors, but I cannot say with confidence about the reliability of the conclusions in the article.
Response 2: Thanks for your comments. The comments are all valuable and very helpful for revising and improving our paper.
As reviewer suggested that we deduct the formulas in the full text and confirm that the conclusions are correct. Besides, we add a statistical error in the table2 and specific analysis based on Figure 5 in section 2.3 which is marked in blue in the paper.
Thank you again for your suggestions. We hope to learn more from you.
Round 2
Reviewer 1 Report
The most worrying thing about the paper is first and foremost the presentation of the paper, which I find confusing. Some terms are still not properly defined, and I do not get what is the main punchline of the paper. Namely, what is so significant about the author's 10 qubit RSP protocol? What are the advantages and disadvantages of using this RSP protocol versus other protocols? Without properly comparing this work on RSP to other RSP work, and also the broader problem of state preparation, it is hard to see why this work warrants the reader's attention.
The authors have made some attentions to explain why there is the need for this very specialized form of RSP, but I remain unconvinced as to the need for such a study. More precisely, if this form of RSP saves resources in some way, the authors should define what resources they are talking about, and quantify precisely the savings in the resources.
Page 3 line 89: states are not quantum channels. It is not clear what this sentence means.
Page 3 Line 93,94. what resources are saved?
what computational complexity is reduced?
please be more explicit.
Page 5, etc. The authors keep saying 'might collapse'. Don't use such imprecise language. Show all the possibilities of the measurement.
Eq 23. It is not clear what the state is, because the dots instead the state are ambiguous, and I could interpret them as potentially anything.
Page 11, line 220. j is not the number of Kraus operators. It is a label.
Reviewer 3 Report
The authors have introduced a number of minor changes, none of them changing the main issues raised in my first report. It is still hard to see what are the novelties contained in this manuscript and therefore its motivation. The IBM experimental part is completely pointless, since as I have stated in my first report probability histograms are not enough to determine the state fidelity.
